# CNN-Based Fault Localization Method Using Memory-Updated Patterns for Integration Test in an HiL Environment

**Ki-Yong Choi** and **Jung-Won Lee** *

Department of Electrical and Computer Engineering, Ajou University, Suwon 16499, Korea
* Correspondence: jungwony@ajou.ac.kr; Tel.: +82-31-219-1813

**Abstract:** Automotive electronic components are tested via hardware-in-the-loop (HiL) testing at the unit and integration test stages, according to ISO 26262. It is difficult to obtain debugging information from the HiL test because the simulator runs a black-box test automatically, depending on the scenario in the test script. At this time, debugging information can be obtained in HiL tests, using memory-updated information, without the source code or the debugging tool. However, this method does not know when the fault occurred, and it is difficult to select the starting point of debugging if the execution flow of the software is not known. In this paper, we propose a fault-localization method using a pattern in which each memory address is updated in the HiL test. Via a sequential pattern-mining algorithm in the memory-updated information of the transferred unit tests, memory-updated patterns are extracted, and the system learns using a convolutional neural network. Applying the learned pattern in the memory-updated information of the integration test can determine the fault point from the normal pattern. The point of departure from the normal pattern is highlighted as a fault-occurrence time, and updated addresses are presented as fault candidates. We applied the proposed method to an HiL test of an OSEK/VDX-based electronic control unit. Through fault-injection testing, we could find the cause of faults by checking the average memory address of 3.28%, and we could present the point of fault occurrence with an average accuracy of 80%.

**Keywords:** automotive software; fault localization; hardware-in-the-loop (HiL); sequential pattern mining; convolutional neural network (CNN)

## 1. Introduction

In recent years, electronic control units (ECUs) and software have been developed in a distributed manner because functions, such as driving assistants and passenger convenience functions, have diversified [1]. The functions developed in this distributed environment are integrated into the system as a black box [2]. An integrated system becomes more complex as the number of components increases, because of dynamic interactions and inter-component reuse [3]. The verification of automotive software has become a key process in ECU development, closely related to safety and performance [4]. The automotive industry has established "Road vehicles—Functional safety (ISO 26262)" as the verification of automotive software becomes increasingly important [5]. ISO 26262 proposes a V-Model and a hardware-in-the-loop (HiL) test for the quality assurance of key components related to safety in automobiles, and divides them into several stages, including unit tests and integration tests [6]. The HiL test is a black-box test that is executed automatically by the simulator in the scenario of the test script. It compares the output value with the expected value, and reports the result only as pass/fail [7].

That is, the reason for any failure is not provided. Therefore, it is necessary to study a means to find the cause of faults that occurred during the HiL test.

In general, developers can use existing debugging tools or software fault-localization methods to debug faults in the development of embedded systems, including ECUs. However, existing debugging tools have the following problems for application to HiL environments. First, the in-circuit emulator (ICE, e.g., Trace32, Multi-ICE), which is widely used as a debugging tool, requires a dedicated connector as a debugging interface. However, the system under test (SUT) of the HiL environment targets rarely exposes the debugging interface to the outside [8]. Second, existing debugging tools should be monitored step by step, using breakpoints. However, because the HiL test is performed automatically by the simulator, it is difficult to suspend the system [7]. In addition, many software fault-localization methods have been developed to detect faults efficiently in the white-box state based on the source code [9]. Therefore, the HiL test, a black box test, needs a method that can be applied without the source code.

A typical method for finding the cause of a fault without any source code is a method using a test case. First, some studies have prioritized test cases using statistics and have detected faults via the spectrum-based fault-localization method [10,11]. In addition, a study determined the attribute that affects the fault by extracting rules based on the results of the test cases [12]. However, because the HiL test evaluates the actual operation result in the hardware, the test case must be generated considering the reaction time [4]. Therefore, the method using the test case needs to consider the time constraints for application to the HiL test.

On the other hand, the cause of the fault can be found in the HiL environment by using memory. In the data-flow analysis, the data is defined and then used. Thus, depending on the definition-use (DU) chain, the footprints of all of the critical data processed by the central processing unit (CPU) while the software is running remain in memory. According to the coding rule of MISRA-C (MISRA-C: 2004 Rule 18.3, 20.4), which is a programming rule used in automotive systems, ECU memory is prohibited from reuse or from dynamic memory allocation [13]. In other words, the variables in the automotive software are statically assigned to the memory of the ECU, and the footprints of the software execution remain. Therefore, the execution history of the software can be checked by observing the memory periodically, and therefore the cause of the fault can be found. For this reason, we have developed a fault-localization method that uses the difference in memory-updated information between unit tests and integration tests [14]. The cause of the fault can be found by comparing the updated addresses in each unit test required for the normal operation of the integrated unit functions of the integration test. However, this method cannot specify when a fault occurs, and when the number of functions to be combined increases or becomes complicated, the major addresses of the executed functions become fault candidates, increasing the number of addresses to be checked. In addition, because this method does not know when the fault occurred, the interpretation of the results requires an understanding of the software's execution flow and a background knowledge of the address symbols and the stored values. Therefore, a method suitable for the HiL environment is needed that can identify the time of occurrence and cause of the fault without any special knowledge.

In this paper, we propose a fault-localization method that uses updated patterns of memory in integrated testing. First, in the preparation step, the address, symbol, stored value and update frequency of the memory used in the normal unit test are analyzed to generate memory-updated information that is necessary for a normal operation of the function. The sequential pattern-mining algorithm [15] is applied to the generated memory-updated information to extract the memory-updated pattern for normal operation. The updated features of the collected memory are learned by a machine-learning model, comprising a convolutional neural network (CNN), using memory-updated information and an updated pattern. Next, in the integration test, the updated feature is classified in the memory data of each frame, and the fault-occurrence timing is found by pattern matching with the updated pattern of the normal operation.

Figure 1 shows how the point of failure is found. In the integration test, a test script is constructed so that function A (four sequences) and B (five sequences) are successively called, such that the updated patterns of A and B appear in order. Therefore, if the integration test is executed normally,

the memory-updated pattern should show the updated patterns of the two functions in order. However, in the memory-updated pattern of the failed integration test, A was found in order for all sequences, but seq4 and seq5 of B were repeated from frame #n, where B is called. The updated patterns of B have an abnormal pattern that is different from the normal operation at frame #n, and it can be seen that a fault occurs in frame #n. Therefore, if our debugging starts from addresses that should be updated to seq4 of B in frame #n, the cause can be found more quickly.

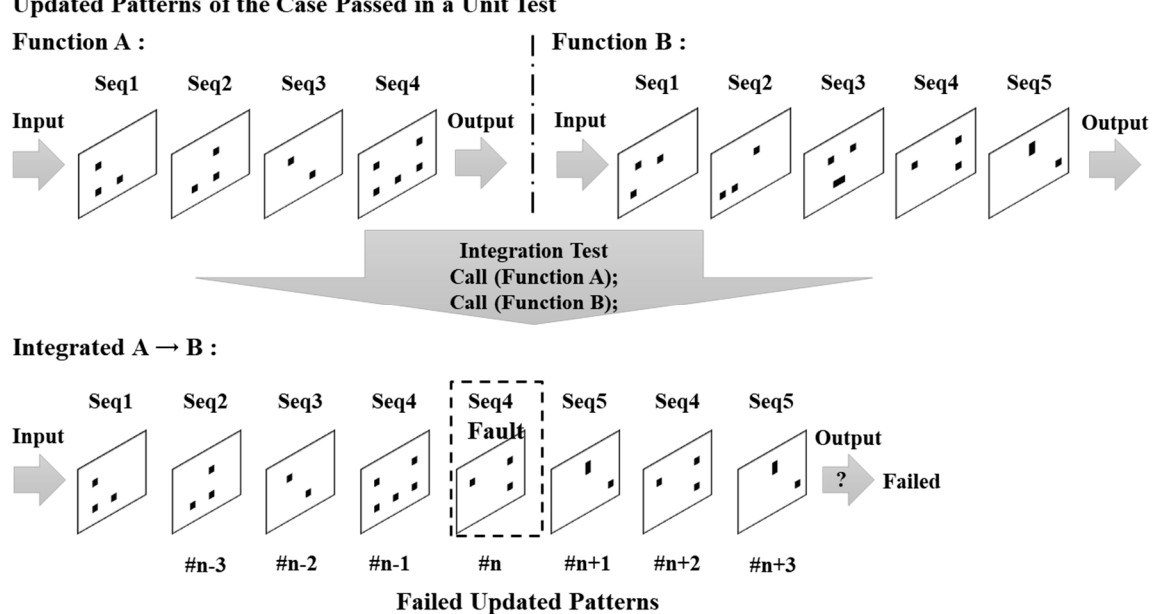

**Figure 1.** Example of fault localization of using memory-updated patterns in an integration test.

As a result, the proposed method can find the occurrence point and the cause of the fault in the integration test by using the normal operation acquired in the unit test, without the source code or the debugging tool in the HiL environment. The proposed method is applied to the injected faults by the mutation method for evaluation in the HiL test of OSEK/VDX-based ECU/SW. The experimental results show the point of occurrence within an average of two frames for 80% faults. Only 3.28% of the memory addresses in the used memory are identified, and the cause of the fault can be found. This result shows that faults are found even if only 57% of the memory is checked by focusing on the fault candidates compared to previous studies on the same environment. The integration test can also indicate when a fault occurred without any special background.

This paper is composed as follows. Section 2 describes related work in debugging in HiL environments and fault-localization methods using sequential pattern mining and CNN. Section 3 defines memory-updated features and pattern extraction for normal operation in memory. In Section 4, we propose a fault-localization method based on the learning method of the updated feature classifier and matching with the normal updated pattern. In Section 5, we configure the HiL environment and test the fault-localization method by injecting faults. Finally, Section 6 concludes with future work directions and conclusions.

## 2. Related Work

The automotive industry conducts HiL testing at the unit and integration test stages for electronic components developed in accordance with ISO 26262. In the HiL test, which is performed automatically without the source code, the simulator evaluates the output of the SUT for the injected input as pass/fail, making it difficult to obtain debugging information when a fault occurs.

In this section, we discuss debugging methods in HiL testing for automotive software, and discuss sequential pattern mining and CNN-based fault-localization methods.

### 2.1. Debugging in an HiL Environment

Automotive software is affected by hardware problems as well as software problems. Therefore, the software must evaluate its behavior on real hardware. The HiL test is a method of evaluating the hardware on which the software is installed [16]. Figure 2 shows the HiL test environment. When a test script is input from the host PC to the HiL simulator, the HiL simulator inputs a signal to the SUT based on the test script. The HiL simulator compares the output from the SUT to the expected value of the test script, evaluates the result as a pass or fail, and reports to the host PC. In this flow, it is difficult for the host PC to know exactly what is happening in the SUT. In other words, the output of the SUT can be checked upon failure, but the internal operation of the SUT is unknown. Therefore, an additional effort is needed to know the internal operation in order to find the cause of the fault.

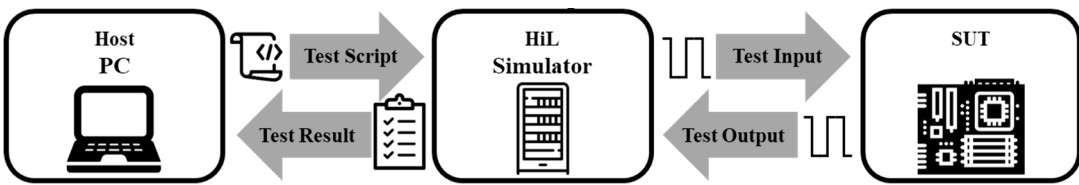

**Figure 2.** Hardware-in-the-loop (HiL) test environment.

Generally, in-circuit emulators (ICEs) and logic analyzers used for debugging embedded systems are difficult to apply to HiL testing. Debugging using ICE, such as Trace32 and Multi-ICE, requires dedicated connectors as a whole system synchronization and debugging interface, to control the system at the instruction level [8]. However, because the HiL test is performed automatically by the simulator, it is difficult to control at the instruction level, and the connector for debugging the completed ECU is rarely exposed. The logic analyzer captures the I/O signal, observes the timing relationship between the signals, and detects the fault signal through the relationship between the signal and the time difference [17]. However, the logic analyzer is suitable for the detection of error signals, but is unsuitable for any internal operation observation, and the tester needs background knowledge at the developer level in order to interpret the signals.

A typical way to find the cause of the fault in a black box test without any source code is to use a test case. Previous studies have prioritized test cases that have a high probability of the occurrence of a fault with statistics that have been detected using a spectrum-based fault-localization method or test history [10,11]. Testing a high-priority test case first reduces the time taken to detect a fault and, as a result, allows for efficient debugging. However, these studies focus on finding test cases that cause faults in black-box testing, rather than finding the cause of the fault. Another study found the rule that causes the fault by analyzing the relation between the test cases where the fault occurred [12]. This study extracted the rule that triggers the fault and finds the attribute that affects the fault. However, this method easily finds the rules when the test cases are simple, but it requires more test cases to extract the rules of the complex input. In other words, the test case-based method requires many test cases to determine the internal operation that causes the actual fault, or to know the attribute related to the fault. The HiL test should account for time constraints because the software evaluates the actual execution results in hardware.

### 2.2. Sequential Pattern Mining in Fault Localization

Sequential pattern-mining algorithms find frequent sequences that can be used to locate associations between multiple items or events in a sequence database [15]. Thus, sequential pattern mining can be used to find associations of events that occur in a time sequence. The fault-localization method using the pattern was mainly performed using discriminative pattern mining. Fault-localization methods that use discriminative pattern mining focus on software behavior patterns. [18] makes use of the fact that the function call tree is different between pass and fail results.

This study examines many test cases to find test cases that are likely to cause a fault in order to give priority to the test cases. The study in [19] obtained a software behavior graph that can show the relationship between a function and a code block. This study provides discriminative nodes for software behavior graphs in pass/fail conditions. In [20], the execution history of the software is tracked to find patterns of repeated events. The fault is located by looking for normal execution and other event patterns. This type of fault-localization study first obtains the normal pattern and uses the difference to locate the fault. Therefore, if a normal execution pattern is acquired in the unit test and an abnormal pattern of the integration test is found, the time when the fault occurs can be determined. However, because it is difficult to know the internal operation information of the HiL test, a method for acquiring an execution pattern is required.

Vehicle software must always be deterministic, because it is closely related to the safety of the passenger. It should always perform the same process and output the same result under the same conditions. If the execution state of the vehicle software is stored at regular intervals, the execution pattern of the software can be extracted using the sequential pattern-mining algorithm. Thus, a periodic collection of ECU memory during the HiL test can be used to extract a sequential pattern of execution traces of the remaining software in memory. This can be used to locate the faults that occur in the integration test if the normal operation pattern of the vehicle software that was operating normally in the unit test is extracted.

### 2.3. Fault Localization Using Convolutional Neural Network

CNN is a type of multilayer neural network whose main feature is recognizing visual patterns directly from pixel images through a minimal preprocessing process. It is less affected by human intervention and background knowledge, because the network learns its own patterns. Some studies have found fault conditions by learning data collected from embedded systems. These studies mainly extract feature points of the input data using 1D CNN. The purpose of [21] is to find bearing faults by inputting motor current signals. The fault is found by learning the current signal when the motor bearing is normal and faulty. In contrast, [22] finds a faulty gear using the vibration of the gearbox when a problem occurs in the gear by learning the vibrations of several healthy conditions, including the fault condition. [23] also finds the faulty gear when a problem occurs in the gear, but this by learning the noise from four types of gearboxes, including the fault condition. However, most of these studies are based upon learning raw data values from existing faults to determine the presence of faults, and cannot provide the debugging time or the causes of faults in integrated tests.

To understand the cause of the fault in the integration test of the HiL environment in a black box, it is necessary to study the following conditions. First, the method should be applicable to HiL environments, such as physical constraints, without source code or debugging tools. Second, it should be able to extract the execution pattern through traces of software execution. Finally, it is necessary to recognize the occurrence of a fault by the observing the memory information of the normal operation, as well as to find the occurrence time and the cause of the fault.

## 3. Data Preparations for Fault Localization

In this chapter, we introduce the necessary preparations for fault localization using the features and patterns that update each address in memory for software execution. First, the data-preparation process is defined, followed by a description of the memory data-acquisition method and memory-updated information analysis method. Next, the mining of the memory-updated pattern and the annotation method of memory-updated features for training are introduced.

### 3.1. Data-Preparation Process

Figure 3 summarizes the preparation process of the fault localization using the training of the memory-updated pattern. To acquire and analyze data periodically and use it for fault localization,

the process is divided into four vertical steps: (1) Data acquisition, (2) updated information analysis, (3) updated pattern detection, and (4) memory transformation for imaging and annotation.

Furthermore, two horizontal phases are necessary for class annotation based on ground truth for training: (1) The unit test phase and (2) the integration test phase. First, the common steps periodically collect memory during the HiL test, analyze the memory data, and generate memory-updated information. From the next steps, there are differences according to test phase. At the unit test step, which collects normal operation information, the memory-updated pattern is found by applying the sequential pattern-mining algorithm into memory-updated information on each frame. CNNs facilitate automatic feature extraction in unit tests, where the training is performed, which enables an exploitation of the ability to identify and distinguish the features of each frame in the integration test, which is an inference phase. Thus, for convenience of use as the input data of CNN, the memory-updated pattern is transformed into an image. In other words, the image presents the total memory in matrix form by mapping a pixel of the matrix to an address and defining it as a frame. As shown in the figure, updated (U) is determined by whether the value stored in each address is changed for each frame acquired in each collection period. This is then transformed into image files, which represent whether the address of each frame has been updated as a marked pixel (updated) or a blank pixel (not updated). The transformed image file is labeled by class, which is the updated feature, as it meets the sequence of the generated updated patterns. Likewise, the integration test, which is the target that is finding the fault, transforms the memory-updated information of the frame into the image file.

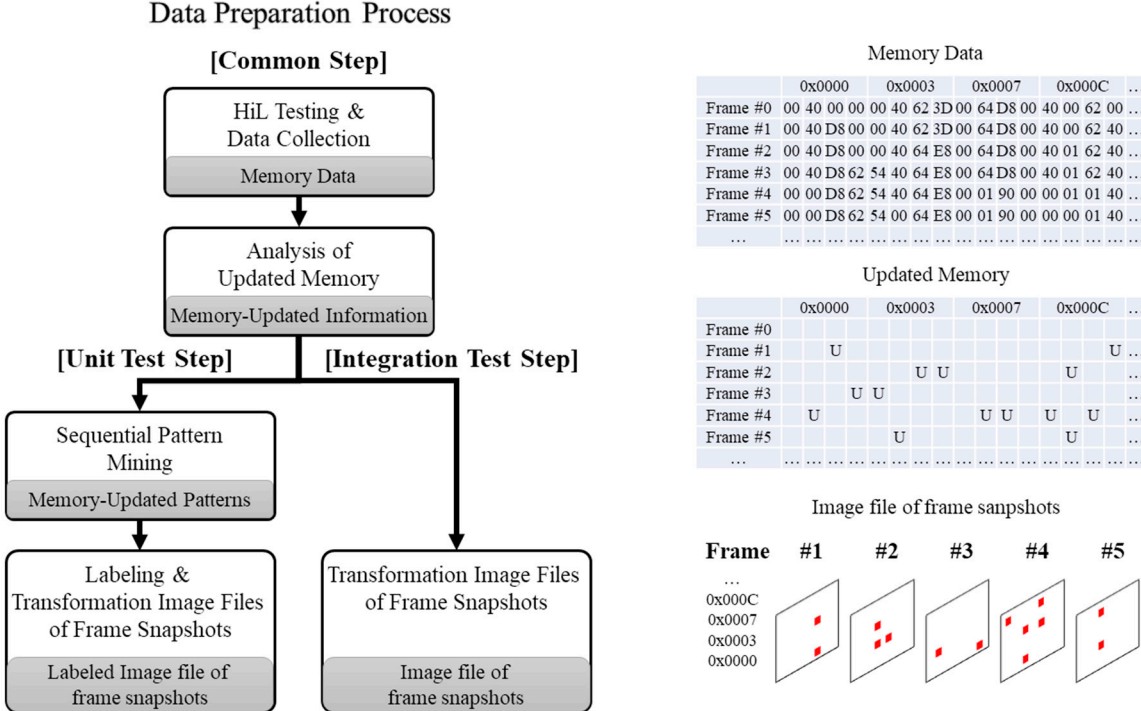

**Figure 3.** Process of data preparation for fault localization using memory-updated patterns.

### 3.2. HiL Testing and Data Collection

The memory-updated feature can be offered to debug faults during HiL testing of automotive software [14]. To generate memory-updated information, the following three types of data are necessary. First, the metadata of tests, such as I/O specifications and test results, are collected through test reports or scripts. Second, the executable file of the software is statically analyzed to collect the symbol name or section information of the memory data to be collected. Third, based on the section information obtained through the static analysis, the memory data is collected for every execution period of the main task.

Collecting executed software memory at every execution period of the main task in an HiL test requires additional space. However, it is impossible to alter the hardware for testing. Therefore, we developed a high-volume data-collecting method considering the communicative load of ECU in HiL environments, as shown in Figure 4. In Figure 4, the test executor of the host PC processes the test script and manages the test result. The data collector requests the memory data from the SUT (test agent) and determines whether the collected memory data is updated. The test agent transmits the memory data of the observation range in the SUT during test time, in accordance to the request. The data collector and the test agent use an automotive communication network, such as a controller area network (CAN).

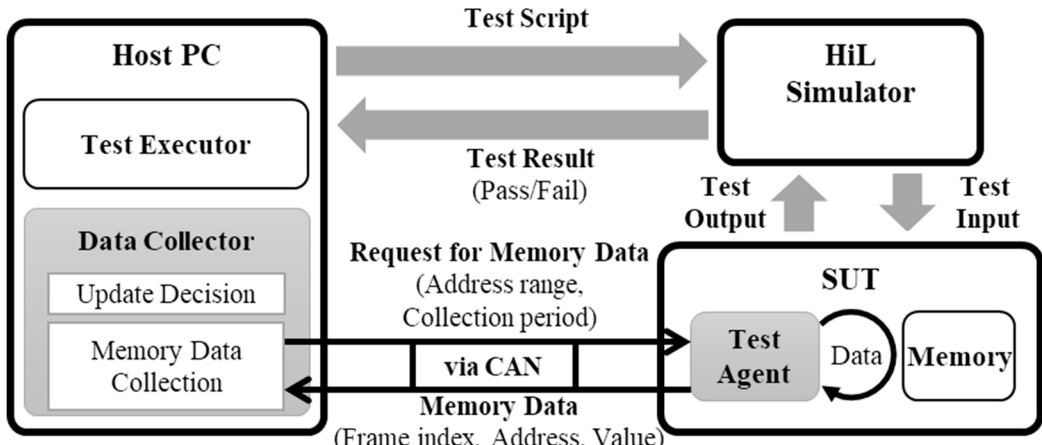

**Figure 4.** Data collector in HiL test environment.

With this method, the memory value of the variables section can be dumped without any loss, and periodically. Dumped memory at the period (T) within the main task is defined as a frame and is presented by Equation (1). The memory address is compared with the prior frame as Equation (2), and checks the updated with the change in the value saved in the address (memory updated: MU).

$$F_k \equiv \{\forall V_{A,k} \,|\, V_{A,k} \text{ is the value of the address A at time } T \times k; k \text{ is an index number}\} \tag{1}$$

$$MU_{A,k} = \begin{cases} 1, & V_{A,k-1} \neq V_{A,k} \\ 0, & V_{A,k-1} = V_{A,k} \end{cases} ; k \text{ is a frame-index number} \tag{2}$$

### 3.3. Analysis of Updated Memory

Synthesizing the collected memory data and analyzing the updated status enables the constitution of memory-updated information (MUI). The MUI is composed of an address (A), a memory-updated frequency ($MUF_{A,R}$) indicating the number of times the value is updated within the set frame range, the value set of the address at each frame ($V_{A,k}$) and the symbol name of the address ($Sym_A$).

$$MUI_{A,R} \equiv \{A, MUF_{A,R}, \{V_{A,k}\}, Sym_A | R \text{ is a set of the frame index}\} \tag{3}$$

Analyzing the composed MUI enables addresses to be found, which is necessary to operate the function normally. By checking the number of inputs and the time interval of the test metadata, the address is found at which the updated interval has the same value as the time interval among the addresses of smaller updates than the number of inputs. In other words, the address updated with an equal time interval of the input is sought. This address is defined as an input-driven updated address (IDUA). The IDUA is updated by the loaded input for testing the functions, and is also necessary for normal operation. Because automotive software should operate deterministically under

identical conditions, it outputs a constant result when performing the determined process. Therefore, IDUA updated by input is updated with a constant pattern.

To obtain the memory-updated pattern of the input, this IDUA that is updated after the input is loaded, and the input-driven updated range (IDUR) is distinguished, which is the frame interval until the result is output. Applying the sequential pattern-mining algorithm for this IDUR enables the extraction of the memory-updated pattern of the functional operation.

### 3.4. Finding Memory-Updated Patterns Using Sequential Pattern-Mining Algorithm

Automotive software is divided into application software, which executes functions, and infrastructure software, which aids in execution [24]. At the IDUR, the addresses of the infrastructure software, such as the support function of the address, or the function related to the kernel to maintain the OS, along with the IDUA related to the function, are updated. Owing to the necessity of the infrastructure's help in operation, the proposed method should look for the update pattern among all updated addresses in the IDUR. Thus, applying the sequential pattern-mining algorithm [25] to this IDUR in a normally operating unit test makes it possible to seek the updated patterns of the normal operation of a function. We modified the open-source code from [26] for application in memory-updated information. In the memory-updated information of the IDUR, each IDUA defines the updated frame as the event, and defines the updated addresses as items at the event. Additionally, it lists events in the sequence of the frame and defines those as a single sequence. Figure 5 shows an example of searching for the updated pattern by applying sequential pattern mining to the memory-updated information.

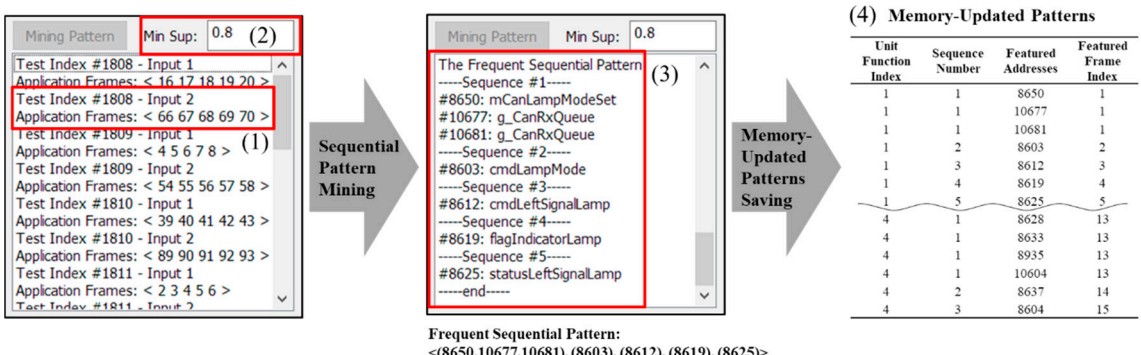

**Figure 5.** Example of sequential pattern mining of memory-updated information.

The example presents an experiment that tests the left signal of the Turn signal. Figure 1 indicates the index for the identifying test and the input number, and the frame number identifying the IDUR according to the input. In the example, in the second input of the 1808th unit test, the IDUR is five frames from 66 to 70. In the case of repeated experiments of test cases with different unit functions, each IDUR appears as many times as there are frames according to which the software operates. Applying the sequential pattern-mining algorithm with minimum support 0.8 in (2) to the updated address of these frames yields the result shown in (3). In (3), the updated addresses are shown in each sequence of the most frequent sequential pattern and symbol name. These addresses are defined as the featured addresses of each sequence. Addresses #10677 and #10681 use the symbol names g_CanRxQueue, unlike other addresses. These addresses are the addresses of the infrastructure software with CAN-related addresses. In the case of normal operation, this software is necessary for the application and infrastructure software. Accordingly, the address of infrastructure is also saved in the memory-updated pattern. In the same way, the updated patterns of the addresses required for all unit functions are extracted. The frequent sequential pattern discovered in subfigure (3) is defined as the featured addresses of each sequence, index of the tested unit function and featured frame index that distinguishes the features of the frame with the sequence number, and saved as (4). In other words,

the Featured Frame Index is defined as a set of {Unit Function index, number of Sequences in the Unit Function index, Featured Addresses}.

### 3.5. Transformation Memory-Updated Patterns to Image Data and Labeling Memory-Updated Features

CNN, which is a multilayer neural network in machine learning, has an advantage in that it directly recognizes the visual pattern of pixel images though minimum preprocessing. As the networks trains directly, it is operated by humans, but not affected by background knowledge. Therefore, using CNN, the updated feature can be automatically extracted from the learning process of the unit test without the intervention of the tester. In this way, it is possible to distinguish the class by identifying the updated feature in the inference process of the integration test. The memory-updated information is transformed into a snapshot, corresponding to a pixel in the image file, in order to be used more easily as an input to the CNN. The size of the transformed snapshot from the memory-updated information in each frame is defined as the height and width obtained by Equation (4), where the coordinate (x, y) of a point in the snapshot is mapped with each address, as in Equation (5). Here, the width is adjusted flexibly as it is in accordance with the collected memory size.

$$\text{Height} = \min\{h \in N; h \geq (\text{Number of Addresses} \div \text{Width})\} \tag{4}$$

$$\text{Address} = y \times \text{Width} + x; (x, y) \text{ is the pixel position} \tag{5}$$

Figure 6a shows an example of transform the memory-updated to snapshot. In the figure, the image file of snapshot has width 160 and height 184. From Equation (2), the memory-updated status of 29,440 addresses is marked as a red pixel when $MU_{A,k}$ has a value of 1 and white when it has a value of 0. Because the coordinates of the marked points are (31,117), the address of the point is 18,751 (i.e., $160 \times 117 + 31$). In Figure 6b, Test Index #1808 has IDUR frames 66–70 in Figure 5, and it generates the memory-updated status of each frame into the image file. As in the figure, the memory data of all of the frames is transformed into image files, and the transformed image file is used for the training phase of the unit test and as well as to detect the occurrence point of faults using a trained classifier in the integration test.

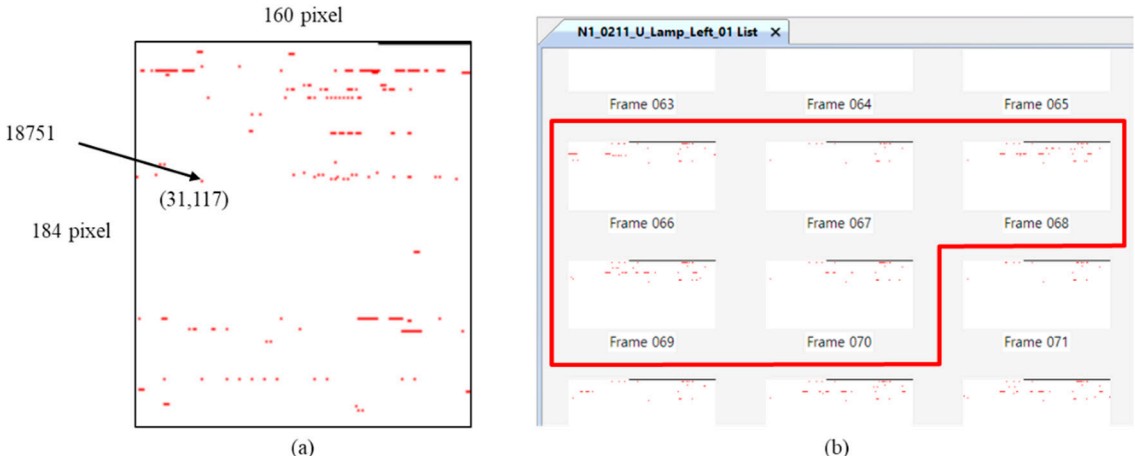

**Figure 6.** Example of transforming memory-updated information into image files. (**a**) Example of a memory-updated snapshot; (**b**) transforming each frame into image files.

The generated image files in unit test are labeled by class for training. First, because the frames 66–70 are classified as the application frame (IDUR) in Figure 5, as Equation (6) it is labeled class as application (App) and the other frames are infrastructure (Infra) class. The updated addresses of frames labeled as application are compared to the featured address of Figure 5(4) to check the unit function index and featured frame index. By Equations (7) and (8), find the updated address of the application

frame in the featured addresses of each index as a unit function and featured frame and label the corresponding index number as class. The labeled image files are classified into the infrastructure, or have a unit function label and featured frame label in the application. The files are also used for the training data of the updated feature-classification model.

$$\text{Class}_{\text{App},k} = \left\{ \begin{array}{ll} \text{App}, & k \in \text{IDUR} \\ \text{Infra}, & k \notin \text{IDUR} \end{array} \right. ; \text{ k is a frame-index number} \tag{6}$$

$$\text{Class}_{\text{Function},k} = \#\text{idx}; \text{ MU}_{\alpha,k} = 1, \ \alpha \in \{\text{set of the featured addresses in unit function } \#\text{idx}\} \tag{7}$$

$$\text{Class}_{\text{Feature},k} = \#\text{idx}; \text{ MU}_{\alpha,k} = 1, \ \alpha \in \{\text{set of the featured addresses in featured frame } \#\text{idx}\} \tag{8}$$

## 4. Fault-Localization Method Using Memory-Updated Patterns

Using the memory-updated features and the memory-updated pattern of the normal operation obtained in the passed unit test, the cause of the fault can be found in the memory-updated information of the failed integration test. In this section, we describe the processes of the fault-localization method and the details of each step.

### 4.1. Fault-Localization Process

An ECU that has passed a unit test performs an integration test under various conditions. Therefore, when a fault occurs in the integration test, it means that the function that operates normally in the unit test is integrated and operates abnormally. In this paper, we use the memory-updated feature and the memory-updated patterns of the ECU that are operated normally in the unit test to find the faulty pattern of the memory-updated information of the faulty integrated test and present it as the fault occurred point. Addresses to be updated at that time are presented as fault candidates.

Figure 7 summarizes the process of localizing the location and occurrence of a fault with the memory-updated patterns prepared in Section 3 and the image of each frame. The process of the proposed fault-localization method is divided into a unit test phase and an integrated test phase. First, in the unit test phase, the updated feature classification model is learned by using the normal updated pattern prepared in the data preparation and the labeled image file. Next, in the integrated test phase, the updated feature classifier is used to classify the image files prepared previously according to the memory-updated feature. The classified frame is compared to the memory-updated pattern of normal operation to determine the sequence of the suspected fault that is not matched. This sequence frame and the previous sequence are specified as fault-occurrence points, and the addresses to be updated in the corresponding sequence are presented as fault candidates.

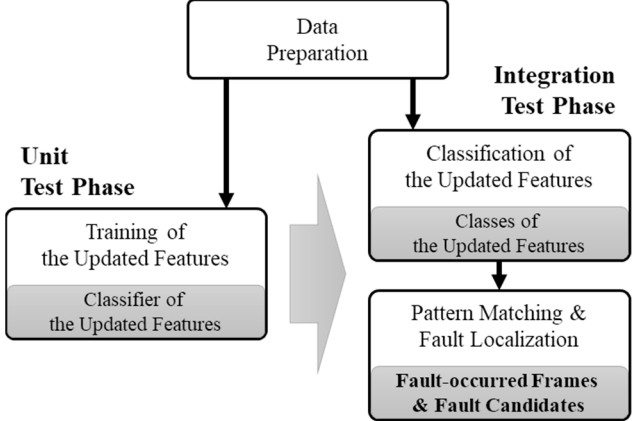

**Figure 7.** Process of fault localization using memory-updated patterns.

## 4.2. Training of Updated Features

The data preparation step is performed on the memory data normally operated in the unit test, and the image file having the class of the updated feature, such as the unit function and the featured frame, together with the normal updated pattern, is generated. In the next step, we design an updated feature classifier that learns the labeled image file and learns the updated feature. At this time, the neural network design should consider the characteristics of the memory-updated snapshot.

A neural network model using CNN is effective for the recognition of neighboring features, while maintaining the spatial information of the image, using a convolution layer as a 2D filter and a pooling layer for feature enhancement [27]. However, the memory-updated snapshots of automotive software have different characteristics from typical 2D images, because each pixel represents an address.

When building automotive software, it is divided into sections according to its purpose, and the address to be allocated to memory is determined according to the object file or function. Therefore, most nearby addresses are used in the same function or belong to the same object file. Because each pixel in the snapshot image corresponds to an address in memory, the nearby addresses are located close to the horizon. A vertically near pixel is actually an address spaced a distance in the horizontal direction, and is less related. Furthermore, at the boundary of the snapshot, the next address is on the other side of the image. As a result, when applying CNN, as in other studies discussed in Section 2.3, the memory-updated snapshots must be converted to 1D for continuity between addresses.

The basic model of the proposed neural network model consists of six layers as shown in Figure 8. First, (1) serializes the input image file and reshapes it into a one-dimensional array. Next, (2) and (3) perform a convolution operation on a one-dimensional filter at stride intervals. At this time, the activation function uses ReLU. ReLU does not saturate the value, but is often used in the CNN by simply activating the value to the threshold [27]. The input data has as many channels as the number of filters in the convolution layer. The channels pass through a flattened layer to convert them into a 1D vector at (4). Then, (5) passes through the fully connected layer and reduces it to the number of features set. Finally, (6) finds the output feature class corresponding to the input frame by applying the softmax function to the fully connected layer. The proposed classifier adjusts the number of filters and the number of features in this basic model.

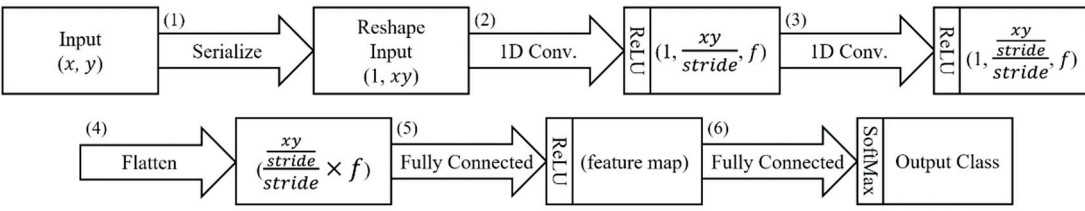

**Figure 8.** The basic form of the proposed neural network model.

The updated feature classifier created using the basic model learns the labeled image file obtained from the unit test. The updated feature classifier consists of three stages. The first stage recognizes whether the application is working for each frame and classifies it into two classes of app/infra. The next stage classifies the unit functions that operate on frames classified as the app class. In the example of Figure 5, if there are four unit function indices, f is set to 4 in steps (2)–(4) in Figure 8, and is learned as a neural network that distinguishes four classes. The last stage classifies the featured frame index. It classifies the featured frame index according to whether each featured address is updated, and classifies the feature of the sequence for the frame confirmed up to the unit function.

## 4.3. Reducing the Fault Candidates

We have described the detailed steps in the unit test phase for fault localization. In the integration test phase, the conversion of the collected memory data to the image file proceeds in the same way as the unit test. The image file contains the updated feature of the unit function and the featured frame.

That is, the updated information of the featured addresses. Apply the proposed fault-localization method to these image files. The proposed method classifies the updated feature for the images of the app class with the traces of the function executed, except for the images of the infra class that are not directly related to the execution of the function. For each frame classified by the app class, the unit function and the featured frame class are identified, and to find a sequence in which a fault occurs that does not match the memory-updated pattern of normal operation.

The updated feature classification uses the updated feature classifier created in the unit test phase. Each frame of the test in which the fault occurred is filtered by application frames corresponding to IDUR through the first stage classifier. Next, application frames classify the unit function and the featured frame in order, and output only the frames whose unit function index and featured frame index match the class. Figure 9 shows the flow through the updated feature classifier. In the figure, the image files of frames 214–221 are classified as the Infra/App class in the first stage. According to the flowchart, files 216, 218, and 219 are classified as the App class, and the matching of the unit function and the featured frame is confirmed. In the example, file 217 is classified as the unit function #4, the featured frame is classified as #5, but the featured frames matching the unit function #4 are #13–15. In this way, the unit function and the featured frame are matched, except for file 217. Finally, files 216, 218, and 219 are classified as application frames, the unit function index is #1, and the featured frame index is output as #1–3.

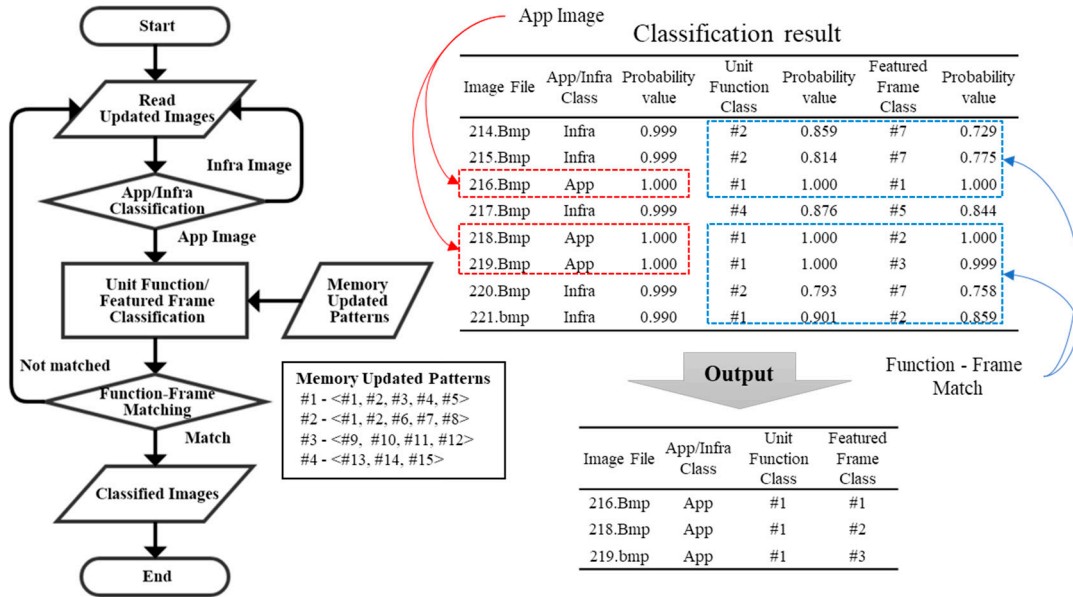

**Figure 9.** Flowchart of classification of updated features.

The frames to which the class is assigned find the sequence corresponding to the updated feature, and compare it with the updated pattern of the normal operation, which presents a sequence other than the normal pattern as a fault occurrence point (i.e., unexpected sequences, nonsequential sequences, and disconnected sequences). The fault occurrence sequence and the previous sequence are presented as R1 and R2 ranks, respectively. In the rank, R1 must be checked first in the sequence in which the abnormal updated pattern occurs, and R2 may be the cause of the abnormal updated in the previous sequence. All sequences performed when a fault occurs may be involved in the fault, and are presented in R3. Addresses that cause the fault can be found by checking sequentially according to the rank. Figure 10 shows an example of the results presented by the fault-localization method proposed in this paper. In Figure 9, only three frames, 216, 218, and 219, are classified as applications. In (a), the sequence corresponding to the unit function index and the featured frame index of each frame, are sequentially from seq1 to seq3 of the front turn signal, and the next, seq4, does not appear. Therefore, R1 at the time of the fault occurrence is seq4, whose sequence is disconnected, R2 is seq3 that is the

previous sequence, and the other sequence is R3. The fault candidates to be presented along with the point of the fault occurrence are the featured addresses of each sequence. The fault candidates are R1 (#8619), R2 (#8612), and R3 (#8603, #8650, #10677, #10681) according to the normal pattern, as shown in (b). Algorithm 1 is described as to how to present a fault-occurrence point.

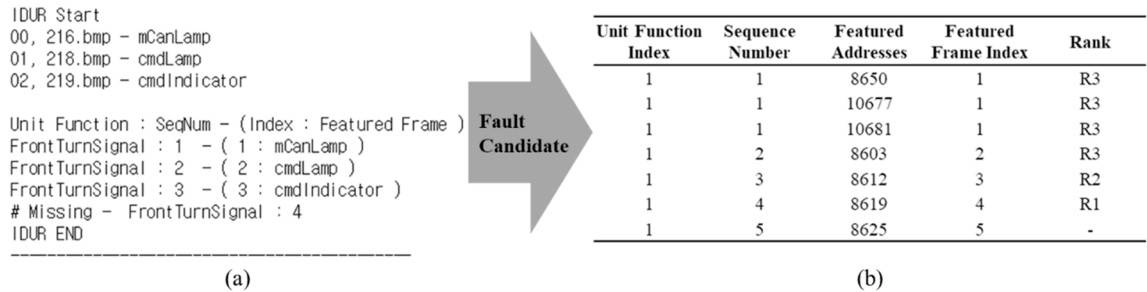

| Unit Function Index | Sequence Number | Featured Addresses | Featured Frame Index | Rank |
|---|---|---|---|---|
| 1 | 1 | 8650 | 1 | R3 |
| 1 | 1 | 10677 | 1 | R3 |
| 1 | 1 | 10681 | 1 | R3 |
| 1 | 2 | 8603 | 2 | R3 |
| 1 | 3 | 8612 | 3 | R2 |
| 1 | 4 | 8619 | 4 | R1 |
| 1 | 5 | 8625 | 5 | - |

(a)  (b)

**Figure 10.** Example of finding the fault occurrence sequence of front turn signal. (**a**) Example; (**b**) fault candidates with the rank.

---

**Algorithm 1.** The algorithm for determining fault occurrence sequence

---

**INPUT**: classified frames, memory-updated patterns
**OUTPUT**: fault occurrence sequence, suspicious sequence
1: PF ≡ The set of unit function in memory-updated patterns
2: S(f) ≡ The set of sequence of PF(f)
3: **FOR** i in classified frames
4:     curPF ← **FIND** (unit function class of i in PF)
5:     **IF** i > 0
6:         **IF** curPF ≠ pastPF
7:             unit_function_change_flag = 1
8:     curSeq ← **FIND** (featured frame class of i in S(curPF))
9:     **IF** I = 0 or unit_function_change_flag = 1
10:         **IF** curSeq ≠ first sequence in S(curPF)                    // *Unexpected sequences*
11:             fault occurrence sequence ← **FIND** (sequence number of curSeq)
12:         **ELSE IF** i<last index of classified frames
13:             **IF** curSeq ≠ next sequence of pastSeq in
S(curPF)                                                                  // *Nonsequential sequences*
14:                 fault occurrence sequence ← **FIND** (sequence number of curSeq)
15:         **ELSE**
16:             **IF** curSeq ≠ last sequence in S(curPF)                 // *Disconnected sequences*
17:                 fault occurrence sequence ← **FIND** (sequence number of curSeq) + 1
18:     pastPF ← curPF, pastSeq ← curSeq
19:     Rank3 ← pastSeq
20: **END FOR**
21: suspicious sequence ← fault occurrence sequence – 1
22: Rank1 ← fault occurrence sequence, Rank2 ← suspicious sequence

---

## 5. Implementation and Experiment

In this chapter, we construct an HiL test environment of OSEK/VDX-based ECU for evaluation, inject a fault for evaluation using a mutation method, and apply the fault-localization method. We injected a fault that was not found in the unit test, but found in the integration test, so that both the normal operation and the fault would be observed in the same software.

*5.1. HiL Test Environment and Fault Injection*

We constructed the SUT as an OSEK/VDX-based software on the MC9S12X [28] ECU for evaluation [14]. Figure 11 shows the HiL test environment configured for the experiment. The environment consists of the SUT with three ECUs, a test executor, and a monitoring system that collects and stores memory data.

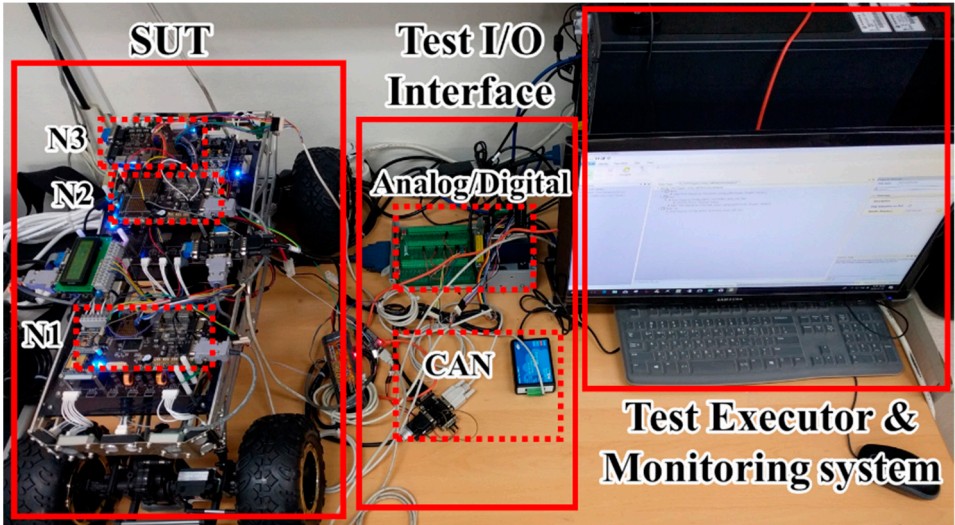

**Figure 11.** HiL test environment.

The SUT has three electronic control units (ECUs) and ten unit functions, and the three ECUs are responsible for the steering and front functions (N1), communication and propulsion systems (N2), and peripheral sensing and rear functions (N3). The software of each unit function is statically assigned to the memory area of each ECU node. Faults injected into the SUT were generated by modifying only one statement in the original source code using the mutation technique [29]. Table 1 summarizes the faults injected into each function.

**Table 1.** List of fault injection by the selected mutation operators.

| Index | Node | Function | Mutation Operator * | Fault |
|-------|------|----------|---------------------|-------|
| 1 | | Front Turn Signal | OEBA | Malfunction due to consecutive commands |
| 2 | N1 | Head Lamp | OCNG | Illumination change error |
| 3 | | Steering | CGSR | Steering angle change error |
| 4 | | AEBS | SSDL | Distance change error |
| 5 | N2 | | CGCR | Operation in the reverse mode |
| 6 | | Driving Mode | STRI | Mode change while accelerating |
| 7 | | | STRI | Increase internal variable in the parking mode |
| 8 | | Rear Turn Signal | OEBA | Malfunction due to consecutive commands |
| 9 | | Brake Lamp | CGSR | Malfunction due to accelerator pedal condition |
| 10 | N3 | Emergency Stop Signal | CGCR | Operation in the reverse mode |
| 11 | | Rear Alert | OEBA | Distance change error |
| 12 | | Adv. ESS | OEAA | Malfunction due to the existing command state |

* OEBA (OEAA): Operator Replacement (Assignment (=) to Bitwise assignment (|=); Assignment to Arithmetic assignment (+ =)) OCNG: Logical context negation CGSR (CGCR): Constant for Scalar (Constant) Replacement SSDL: Statement Deletion STRI: Trap on if Condition.

Because the proposed method operates normally in the unit test and the fault occurred in the integration test, the fault to be injected must also pass the unit test. Therefore, we classify the applicable operators for each function among 80 mutation operators in C language.

Then, the mutation operator that passed the unit test is injected into the fault. At this time, the unit test tests the function on/off in the function initialization state, and in the integration test, the communication between the nodes, the function connection, and the continuous state changes of the single function, are examined. For example, fault #1 modified a statement from the source code of the front turn signal function of Node 1 by using the OEBA operator. This fault has been modified to change an assignment to a bitwise assignment, which works fine in a single command, but the fault occurs when successive commands are received.

## 5.2. Finding the Updated Patterns and Building a Pattern Classifier

The unit test of the SUT implanted with the fault is repeated to generate memory-updated information from the memory data in which the unit functions normally operate, and sequential pattern mining is applied to extract the normal memory-updated pattern. Table 2 shows the memory-updated patterns extracted from the SUT. The functions injected with faults in Table 1 consist of a combination of the unit functions shown in Table 2. For example, the driving mode of node 2 is a combination of the unit function gear and pedal. On the other hand, in the updated pattern of Table 2, the sequence may partially coincide, such as the front turn signal and the head lamp. This is because, even if the unit functions are different, a function similar to the case of allocating a signal on a bit-by-bit basis in the same memory in a communication process, may use a global variable for signal transmission. Both are part of the normal updated patterns. The memory-updated pattern of normal operation in Table 2 indicates the addresses to be updated when each unit function operates and the order to be updated. Therefore, it is possible to judge whether the function is normal by comparing the memory data when the function is operated and the pattern of Table 2.

**Table 2.** List of memory-updated patterns of the system under test (SUT).

| Node | Unit Function | Number of Sequences | Featured Addresses |
|---|---|---|---|
| 1 | Front Turn Signal | 5 | (8650,10677,10681), (8603), (8611,8612,8613), (8619), (8624,8625,8626) |
|  | Head Lamp | 5 | (8650,10677,10681), (8603), (8605,8608,8609), (8615,8610,8617,8618), (8620,8622,8623) |
|  | Steering | 4 | (8651,10677,10681), (8639), (8640), (8641) |
|  | Sensor | 3 | (8628,8633,8935,10604), (8637), (8604) |
| 2 | AEBS | 9 | (8696), (8700,8704), (8668,8669,8704), (8661), (8628), (8701,8703,8704), (8671,8672,8678,8683,8688,8704), (8664), (8624,8627) |
|  | Gear | 4 | (8700,8704), (8668,8669,8704), (8661), (8628) |
|  | Pedal | 4 | (8701,8703,8704), (8671,8672,8673,8683,8688), (8664), (8624,8627) |
| 3 | Rear Turn Signal | 5 | (8646), (8606), (8607,8608,8609), (8621), (8624,8625,8626) |
|  | Brake Lamp | 5 | (8646,8648), (8613,8614), (8610,8611,8619,8620), (8622,8623), (8627,8628) |
|  | Rear Alert | 4 | (8656), (8618), (8646), (8602,8613) |

We generated the data to learn the updated feature using the updated patterns of the normal operation obtained. Based on the update of the featured addresses, the frame of the range in which the function operates in the unit test is converted into an image file and classified into a sequence. Infrastructure frames are sampled by the number of frames classified as applications in each unit test. Anaconda [30] and Keras [31] were used to generate and learn the update feature classifier.

The data set for learning consisted of about 4000 image data. For the featured frame of 48 sequences of memory-updated patterns in Table 2, image data of 40 app class on the average were prepared, and data of infra class of the similar amount were also prepared. These image data contain labels of three classes that are necessary for the learning of the three classifiers. Use this dataset to learn three classifiers for each node, a total of nine classifiers. The dataset was constructed considering the ratio of the featured frame. Therefore, when learning the unit function classifier, the data ratio between classes should be adjusted evenly. Among the data sets, the ratio of training to validation is 4:1. Table 3 shows the accuracy, precision, recall, and F1-score for the validation data as a result of learning. In this case, the F1-score is the harmonic mean of precision and recall.

The nine classifiers learned were 96–100% overall for the four measures, and all four measures were learned to over 98% on average.

**Table 3.** Training results of the classifiers using a convolutional neural network (CNN).

| Node | Classifier | Accuracy (%) | Precision (%) | Recall (%) | F1-Score |
|------|-----------|--------------|---------------|-----------|----------|
| 1 | App/Infra | 96.06 | 96.07 | 96.40 | 96.23 |
| | Unit Function | 97.92 | 98.47 | 98.12 | 98.29 |
| | Featured Frame | 98.90 | 99.13 | 98.70 | 98.92 |
| 2 | App/Infra | 100.00 | 100.00 | 100.00 | 100.00 |
| | Unit Function | 97.99 | 97.96 | 97.82 | 97.89 |
| | Featured Frame | 98.80 | 97.09 | 96.94 | 97.02 |
| 3 | App/Infra | 98.44 | 98.39 | 98.57 | 98.48 |
| | Unit Function | 96.84 | 97.44 | 97.39 | 97.41 |
| | Featured Frame | 98.44 | 97.86 | 98.66 | 98.26 |
| Average (%) | | 98.15 | 98.04 | 98.07 | 98.06 |

*5.3. Experimental Result*

Among the 12 faults injected in Table 1, localization results are presented through Fault index #1 (disconnected sequence case) and Fault index #12 (unexpected sequence case). We compare the present fault-occurrence point with the actual fault-occurrence point and compare the proposed method with the localization ratio of the previous study.

5.3.1. Finding the Fault in the Disconnected Sequence (Fault Index #1)

This fault is when the input normally enters the function, but the input is not passed to the output. In other words, the updated patterns of the memory according to the operation of the function are not connected to the sequences. In Table 1, fault index #1 is a malfunction due to consecutive commands input to the front turn signal function. That is, when a function is in operation, a command for another operation is input, and a fault occurs. We conducted a test to turn on/off each operation in the unit test. In the integration test, consecutive instructions were input at intervals of 500 ms. The result is shown in Figure 12. In the figure, the IDUR of 166–171 normally shows the pattern, and the IDUR of 216–218 is the fourth sequence, but is missed. Because the frame interval is 10 ms, the 50-frame difference is 500 ms. Therefore, in the following operation command, the previous input operates normally, and the subsequent input shows that a fault has occurred.

According to Algorithm 1, sequence #4 of the front turn signal is the fault-occurrence sequence (R1), #3 is the suspect sequence (R2), and the other sequences are the related sequence (R3). At this time, it can be seen that the proposed method focuses on sequential patterns rather than continuity in 167 and 217 frames that are excluded. As a result, according to Table 2, the fault candidates of the fault index #1 is R1 (#8619), R2 (#8611, #8612, #8613) and R3 (#8650, #10677, #10681, #8603, #8624, #8625, #8626). In fact, the fault we injected is mutated to '=' with '|=' in the assignment operation of the variable assigned to #8619, Rank R1, according to the OEBA operator.

```
IDUR Start                                       IDUR Start
00, 166.bmp — mCanLamp                           00, 216.bmp — mCanLamp
01, 168.bmp — cmdLamp                            01, 218.bmp — cmdLamp
02, 169.bmp — cmdIndicator                       02, 219.bmp — cmdIndicator
03, 170.bmp — flagIndicator
04, 171.bmp — statusIndicator                    Unit Function : SeqNum — (Index : Featured Frame )
                                                 FrontTurnSignal : 1  — ( 1 : mCanLamp )
Unit Function : SeqNum — (Index : Featured Frame )  FrontTurnSignal : 2  — ( 2 : cmdLamp )
FrontTurnSignal : 1  — ( 1 : mCanLamp )          FrontTurnSignal : 3  — ( 3 : cmdIndicator )
FrontTurnSignal : 2  — ( 2 : cmdLamp )           # Missing — FrontTurnSignal : 4
FrontTurnSignal : 3  — ( 3 : cmdIndicator )      IDUR END
FrontTurnSignal : 4  — ( 4 : flagIndicator )     ___________________________________
FrontTurnSignal : 5  — ( 5 : statusIndicator )
IDUR END
___________________________________
```

**Figure 12.** Finding the fault-occurrence point of fault index #1.

### 5.3.2. Finding the Fault in the Unexpected Sequence (Fault Index #12)

This fault is the case when a normal input is sent to a function, but an unexpected result is output. In other words, the memory-updated pattern of the function is a case where an unexpected sequence appears in the normal sequence. In Table 1, fault index #12 indicates a malfunction due to the existing command state. Advanced emergency stop signal (Adv. ESS) is a function that turns on the rear emergency signal in the rear turn signal when it stops completely, in addition to the ESS function that flashes the brake lamp in an emergency braking situation. In the integration test, we made the ESS operate with the turn signal turned on, resulting in a fault of the emergency signal. As a result, we obtained the same result as Figure 13. In the figure, the fourth sequence of the rear turn signal suddenly appears when moving from 202 to 203 frames. Thereafter, a fault occurs in which the fourth and fifth sequences are repeated in the frame. Therefore, sequence #4 of the rear turn signal is the fault-occurrence sequence, #3 is the suspicious sequence (R2), and #5 and the brake lamp sequences are the related sequence (R3). In fact, the fault we injected was mutated to '=' with '+=' in the assignment operation of #8621, Rank R1, according to the OEAA operator.

```
IDUR Start                         Unit Function : SeqNum — (Index : Featured Frame )
00, 198.bmp — mCanPedal            BrakeLamp : 1  — ( 6 : mCanPedal )
01, 199.bmp — cmdPedal             BrakeLamp : 2  — ( 7 : cmdPedal )
02, 200.bmp — cmdBrakeLamp         BrakeLamp : 3  — ( 8 : cmdBrakeLamp )
03, 201.bmp — flagBrakeLamp        BrakeLamp : 4  — ( 9 : flagBrakeLamp )
04, 202.bmp — statusBrakeLamp      BrakeLamp : 5  — ( 10 : statusBrakeLamp )
05, 203.bmp — flagIndicator        RearTurnSignal : 4  — ( 4 : flagIndicator )
06, 205.bmp — flagIndicator        RearTurnSignal : 4  — ( 4 : flagIndicator )
07, 209.bmp — flagIndicator        RearTurnSignal : 4  — ( 4 : flagIndicator )
08, 210.bmp — statusIndicator      RearTurnSignal : 5  — ( 5 : statusIndicator )
09, 213.bmp — flagIndicator        RearTurnSignal : 4  — ( 4 : flagIndicator )
10, 214.bmp — statusIndicator      RearTurnSignal : 5  — ( 5 : statusIndicator )
11, 217.bmp — flagIndicator        RearTurnSignal : 4  — ( 4 : flagIndicator )
12, 218.bmp — statusIndicator      RearTurnSignal : 5  — ( 5 : statusIndicator )
                                   # Suddenly appear — ['RearTurnSignal : 4']
                                   IDUR END
                                   ___________________________________
```

**Figure 13.** Finding the fault-occurrence point of fault index #12.

### 5.3.3. Evaluation

We also applied the proposed method to the other 10 faults in Table 1. As a result, we could find the cause of the fault in the fault candidate that presented for all the faults. The experimental results are summarized in Table 4. Of these, the actual fault was highlighted in the shadow cell at which it was found. As a result, faults were found in R1 and R2 in 6 cases out of 10. We could find the cause of the actual fault within a frame distance of 0.875, on average, at the fault-occurrence point that we proposed. That is, if the frame indicated by the fault-occurrence point and an additional frame are

further checked, the cause of the fault can be found. The localization ratio is obtained by dividing the number of addresses up to the rank where the fault was found by the total number of updated addresses. For example, fault index #9 is (2 + 4)/132 with a 4.55% localization. As a result, our proposed method could find the point where the fault was updated by checking two frames on the average for 80% fault in the experiment to find the injected fault. The fault was found by checking only 3.28% of the total memory addresses updated.

**Table 4.** Experimental result.

| Fault Index | Function | Allocated Memory (Byte) | Updated Addresses (Byte) | Number of Fault Candidates | | | Frame Distance (Frames) | Localization (%) | Method [14] |
|---|---|---|---|---|---|---|---|---|---|
| | | | | R1 | R2 | R3 | | | |
| 1 | Front Turn Signal | | 187 | 1 | 3 | 7 | 0 | 0.53 | 9(4.61%) |
| 2 | Head Lamp | 3714 | 195 | 1 | 1 | 7 | 0 | 0.51 | 14(7.18%) |
| 3 | Steering | | 200 | 1 | 1 | 4 | 1 | 1.00 | 4(2.00%) |
| 4,5 | AEBS | 4032 | 221 | 0 | 0 | 20 | - | 9.05 | 22(9.95%) |
| 6,7 | Driving Mode | | 179 | 0 | 0 | 17 | - | 9.50 | 13(7.26%) |
| 8 | Rear Turn Signal | | 170 | 1 | 3 | 5 | 0 | 0.59 | 9(5.29%) |
| 9 | Brake Lamp | | 132 | 2 | 4 | 6 | 1 | 4.55 | 7(5.30%) |
| 10 | ESS | 3711 | 289 | 2 | 4 | 4 | 4 | 3.46 | 9(3.11%) |
| 11 | Rear Alert | | 167 | 1 | 1 | 3 | 1 | 2.99 | 4(2.40%) |
| 12 | Adv. ESS | | 163 | 1 | 3 | 15 | 0 | 0.61 | 17(10.43%) |
| | Average | | | | | | 0.875 | 3.28% | 5.77% |

On the other hand, our proposed method does not suggest when is the fault-occurrence point in the case of Node 2. The proposed method uses the update pattern to present the point of occurrence of the abnormal pattern as the point of occurrence of the fault. Therefore, we confirmed the limit at which the fault-occurrence point cannot be determined when the updated patterns of the memory are normally maintained and updated to the wrong value. However, this limitation can be overcome. As shown in Table 4, all the detected sequences are presented as fault candidates for Rank R3, so we can find the cause of the fault by localization similar to the previous study.

As a result, the proposed method can reduce the fault candidates strongly by presenting the point of occurrence for 80% of faults and can suggest a fault candidate similar to the previous study, even when the point of occurrence cannot be observed. When we test the same fault, the proposed method is localized to 3.28%, which is 57% of the memory compared to the previous study. We apply the proposed method to ECUs based on OSEK/VDX. This means that our proposed method can find faults that occur in the HiL test environment using the normal memory-updated features and the updated patterns without debugging tools or source code. However, because the proposed method only targets the update, not the value of the memory, it is limited when there is an error in the value while maintaining the normal pattern. In addition, because it focuses on updating, it is difficult to apply analog signals with frequent changes in values.

## 6. Conclusions

The proposed method collects memory data in the cycle of main task execution during the HiL test. Analyzing the updated information of collected memory and applying the sequential pattern-mining algorithm enables the acquisition of both the updated feature and updated memory patterns of software execution. The time of fault occurrence indicates that the frame sequence does not match the updated patterns of normal operation in the frame of the integration test where the fault occurs. The fault candidates are categorized into ranks of the featured addresses of the fault-occurrence sequence and the executed sequence. As a result of the experiment by fault injection, 80% of faults suggest the time of fault, and the fault candidates were localized to 3.28% on average. If 100 variables are used in the integration test, the fault can be found by checking only 3 variables before and after the fault occurred at the time of debugging.

The proposed method has the following advantages. First, fault-occurrence time is suggested in a black-box environment, where the source code is difficult to use and the memory address that causes the fault can also be detected. The proposed method, based on normal operation information, that acquires the unit test, can assess the difference in fault situations. Second, with the only automotive communication interface, it is possible to obtain debugging information, even without using existing debugging tools. The proposed method collects memory in the cycle of the main task period of the system, which has a comparable effect of observing the system without debugging tools. Third, the fault-occurrence time and cause of the fault are detected, based in the memory-updated features and the updated pattern of normal operation. Thus, those who perform the test can apply the proposed method even without any background knowledge about the execution flow of the software or the characteristics of the input and output signals. Therefore, the proposed method is based on the memory-updated features and updated patterns without source code or existing debugging tools, and determines not only the time of fault occurrence, but also the memory address responsible to the fault. The developer can thus reduce the time required for debugging.

As our method considers the updated status, not to the saved value in the memory, in cases in which the saved value in a major address is updated to the wrong value, it is difficult to determine the time of the fault occurrence. Additionally, because we focus on the updated status, it is limited in the case in which the updated status is not important, like an analog signal, which is common. However, in addition to suggesting the time of fault occurrence in discrete signals, like digital I/O, its localization feature is more robust than that in existing works, which only used the memory-updated information. Furthermore, we plan to proceed with further work using the pattern of change in the value stored in memory.

**Author Contributions:** J.-W.L. conceived and designed the experiments; J.-W.L. and K.-Y.C. analyzed the data; K.-Y.C. performed the experiments, contributed analysis tools, and wrote the paper.

**Funding:** National Research Foundation of Korea: NRF-2016R1A2B1014376, MSIT(Ministry of Science and ICT), Korea: IITP-2019-2016-0-00309

**Acknowledgments:** This research was supported by the National Research Foundation of Korea (NRF) grant funded by the MSIP (NRF-2016R1A2B1014376); This research was supported by the MSIT (Ministry of Science and ICT), Korea, under the ITRC (Information Technology Research Center) support program (IITP-2019-2016-0-00309) supervised by the IITP (Institute for Information & communications Technology Planning & Evaluation).

**Conflicts of Interest:** The authors declare no conflict of interest.

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
