# Peer review of "CNN-Based Fault Localization Method Using Memory-Updated Patterns for Integration Test in an HiL Environment"

_applsci, doi:10.3390/app9142799_

Round 1

Reviewer 1 Report

The article entitled CNN-based fault localization method using 2 memory-updated patterns for integration test in an 3 HiL environment is well written and would be interest for the readers of Applied Sciences. In spite of these, I would like to recomed some minor changes before its publication:

From line 313, section "Transformation Memory-Updated Patterns to Image Data and Labeling Memory-Updated Features" a more in Depth mathematical explanation is required.

Refence 3: please check the use of capital letters in the name of authors.

Author Response

We have uploaded, which summarize the answers and revisions.

Reviewer 2 Report

The purpose and results of the paper are demonstrated well. 

Author Response

We sincerely appreciate your valuable time and review.

Reviewer 3 Report

I consider the article interesting, at least for the type of application that is made of CNN. I think it is well structured in the paragraphs and the motivations are clear. Personally, I would improve the exposure of the dataset in section 5, in particular dimension and how is structured. Finally check some small selling errors for the English.

Author Response

We have uploaded, which summarize the answers and revisions.

Appl. Sci. EISSN 2076-3417 Published by MDPI AG, Basel, Switzerland RSS E-Mail Table of Contents Alert
Back to Top